# Alteration of Immunoregulatory Patterns and Survival Advantage of Key Cell Types in Food Allergic Children

**DOI:** 10.3390/cells12232736

**Published:** 2023-11-30

**Authors:** Kamal Ivory, Rossella Angotti, Mario Messina, Denise Bonente, Ferdinando Paternostro, Massimo Gulisano, Claudio Nicoletti

**Affiliations:** 1The Quadram Institute, Norwich NR4 7UQ, UK; kivory2@icloud.com; 2Department of Medicine, Surgery and Neuroscience, University of Siena, 53100 Siena, Italy; rossella.angotti2@unisi.it (R.A.); mario.messina@unisi.it (M.M.); 3Department of Molecular and Developmental Medicine, University of Siena, 53100 Siena, Italy; denise.bonente@student.unisi.it; 4Department of Experimental and Clinical Medicine, University of Florence, 50134 Firenze, Italy; ferdinando.paternostro@unifi.it (F.P.); massimo.gulisano@unifi.it (M.G.)

**Keywords:** food allergy, cytokines, immunoregulation, pediatric allergy, apoptosis

## Abstract

All allergic responses to food indicate the failure of immunological tolerance, but it is unclear why cow’s milk and egg (CME) allergies resolve more readily than reactivity to peanuts (PN). We sought to identify differences between PN and CME allergies through constitutive immune status and responses to cognate and non-cognate food antigens. Children with confirmed allergy to CME (*n* = 6) and PN (*n* = 18) and non-allergic (NA) (*n* = 8) controls were studied. Constitutive secretion of cytokines was tested in plasma and unstimulated mononuclear cell (PBMNC) cultures. Blood dendritic cell (DC) subsets were analyzed alongside changes in phenotypes and soluble molecules in allergen-stimulated MNC cultures with or without cytokine neutralization. We observed that in allergic children, constitutively high plasma levels IL-1β, IL-2, IL-4, IL-5 and IL-10 but less IL-12p70 than in non-allergic children was accompanied by the spontaneous secretion of sCD23, IL-1β, IL-2, IL-4, IL-5, IL-10, IL-12p70, IFN-γ and TNF-α in MNC cultures. Furthermore, blood DC subset counts differed in food allergy. Antigen-presenting cell phenotypic abnormalities were accompanied by higher B and T cell percentages with more Bcl-2 within CD69^+^ subsets. Cells were generally refractory to antigenic stimulation in vitro, but IL-4 neutralization led to CD152 downregulation by CD4^+^ T cells from PN allergic children responding to PN allergens. Canonical discriminant analyses segregated non-allergic and allergic children by their cytokine secretion patterns, revealing differences and areas of overlap between PN and CME allergies. Despite an absence of recent allergen exposure, indication of in vivo activation, in vitro responses independent of challenging antigen and the presence of unusual costimulatory molecules suggest dysregulated immunity in food allergy. Most importantly, higher Bcl-2 content within key effector cells implies survival advantage with the potential to mount abnormal responses that may give rise to the manifestations of allergy. Here, we put forward the hypothesis that the lack of apoptosis of key immune cell types might be central to the development of food allergic reactions.

## 1. Introduction

Maintaining oral tolerance is one of the main tasks of the intestinal immune system that is achieved through cross-talk between the microbiota, epithelium and underlying immune system. Defective homeostasis can result in dysregulated immune responses such as those found in food allergy. There are many implied causal factors, including perturbations in the composition of the intestinal microbiota [1], an imbalance in Th1/Th2 responses [2], deficient T cell [3] or DC [4] regulatory function, changes in antigen presentation [5], abnormalities in the expression of β_7_ integrin [6] and mutations in the filaggrin [7] or cytokine receptor [8] genes. Multiple factors are likely to be involved and the susceptibility of any individual will be determined by any of these, alone or in combination resulting in a failure of oral tolerance. All allergic reactions are recall responses and, like initial sensitization events, their effective regulation requires T-cell and antigen-presenting cell (APC) interactions involving the CD28/CD152-CD80/CD86 signaling pathway [9].

The CD28/CD152-CD80/CD86 signaling pathway is one of the most important regulators of both Th1 and Th2 responses in vivo [10]. It is essential for the development of allergic reactions to food antigens and crucial for inducing tolerance to them [11]. CD28 and CD86 are largely constitutive on the surface of respective T and antigen-presenting cells, whereas CD152 and CD80 are upregulated on activation [12]. CD80 interaction with CD152 is crucial for tempering the intensity of initial sensitization events and also the induction of low dose tolerance [12,13]. While the mechanisms by which this occurs are not known, ligation of CD80 on APCs with constitutively expressed CD152 on regulatory T cells (Treg) may activate the indoleamine-2,3-dioxygenase pathway in DCs, leading to the production of immunosuppressive kynurenin [14]. As CD86 favors CD23-CD21 pairing, it consequently functions as a selective and potent costimulus for IgE synthesis by activated B cells [15]. CD23 is the low affinity receptor for IgE. It exists in membrane-bound and soluble forms and participates in both the positive and negative regulation of IgE synthesis by B and other cell types [16,17,18]. As a multifunctional molecule, CD23 can also promote B cell survival via receptor-mediated inhibition of apoptosis [19] or through the upregulation of Bcl-2 [20,21]. Like B cells, T and other cells also require ongoing signals to maintain their viability, and in an allergic setting, IL-4 can upregulate Bcl-2 [22,23]. Such deregulation of cell death in key cells is likely to have a significant effect on allergic responses as previously observed in mice [24,25].

With this in mind, we have documented the expression of these key molecules, as well as sCD23, Bcl-2 and the early activation marker CD69. We have also quantified DC subsets and measured cytokines released spontaneously or in response to antigenic stimulation as these proteins are central to signaling events in any immune response. Some food allergies resolve spontaneously (e.g., CME), while others are less likely to do so (e.g., PN) [26]; thus, it is clear that, alongside shared features of immune induction/dysregulation, there must also be characteristics that are determined by the nature of the allergen itself in the genetic and immunological context of the individual involved. Our study therefore sought to document phenotypic and functional similarities and differences between PN and CME allergic children, and to compare them with profiles derived for those without any allergies.

## 2. Methods

### 2.1. Study Population

Children with PN (eighteen, age range 3–12 years of age, average 9.6) or CME (six, age range 4–13, average 10.2) allergies attending pediatric allergy clinics alongside NA (eight, age range 4–10, average 8.7) children were invited to participate in the study. Children with additional allergies were excluded from the study. As stated, only children mono-allergic to PN were included or those with CME without additional allergies. Non-allergic children had no known allergies. The diet was unknown, but at the time of study, all participants avoided foods with offending allergens. Only allergic patients with confirmed mono-allergic status were enrolled in the study. This was determined by using an SPT-induced wheal size of 8 mm or more and an allergen-specific IgE level of 15 KUA/L or greater. Total IgE levels were not determined as these were deemed to be poorly specific and may have given false positive sensitization results. The parents of all children signed an informed consent form approved by the local Ethics Committee; LREC No: 361/01.

### 2.2. Allergens

The preparation of crude peanut (*Arachis hypogea*) extract was carried out as described previously [27]. Stock solution was at 100 mg/mL, and it was used at 10 mg/mL culture. Egg albumin (Cat. A5503), milk casein (Cat. C3400) and β-lactoglobulin (Cat. L0130) were purchased from Sigma-Aldrich (Gillingham, UK), stored at 10 mg/mL stock solutions and used at 10 µg/mL culture.

### 2.3. Plasma Collection and Isolation and Culture of Peripheral Blood Mononuclear Cells

Blood was collected into sodium heparin vacutainer tubes (Becton Dickinson, Oxford, UK) at the relevant center and sent by courier at ambient temperature. On receipt, the blood was centrifuged at 700× *g* for 10 min, plasma was collected and frozen at −80 °C until use. Peripheral blood mononuclear cells (PBMNCs) with a viability >95% were prepared as described previously [28]. Then, 1 × 10^6^ MNCs were cultured in the absence or presence of allergen and with or without neutralizing antibodies added at culture initiation. After 6 days at 37 °C in an atmosphere of 5% CO_2_ in air, culture supernatants were removed, centrifuged at 380× *g* for 5 min and frozen at −80 °C until use. Cells were harvested, washed in phosphate-buffered saline containing 0.1% sodium azide, 2% fetal bovine serum and 4 mM EDTA (PBSAAE) with centrifugation at 380× *g* for 5 min. Washed cells were stained as described below.

### 2.4. Antibodies

Neutralizing antibodies anti-TNF-α (Cat. AB-210-NA), pan-specific TGF-β (Cat. AB-100-NA), anti-IL-6 (AB-206-NA) and anti-IL-10 (Cat. AB-217-NA) were purchased from RnD Systems (Abingdon, UK). Anti-IL-4 (Cat. 554481) was purchased from BD Biosciences (Wokingham, UK). Each was used at 10µg/mL culture. Furthermore, the antibodies for cell phenotyping and the appropriate isotype controls were as follows: HLA-DR-ECD (Cat. PN IM3636), CD19-ECD (Cat. A07770), CD4-ECD (Cat. 6604727) and IgG1-ECD (Cat. A07797), which were purchased from Beckman Coulter (High Wycombe, UK). The remaining antibodies were purchased from BD Biosciences (UK): CD80-FITC (Cat. 557226), CD83-FITC (Cat. 556910), CD86-FITC (Cat. 555657), CD3-FITC (Cat. 561807), CD83-PE (Cat. 556855), CD152-PE (Cat. 555853), CD86-APC (Cat. 555660), CD69-APC (Cat. 555533), anti-Bcl-2-PE set (Cat. 556536), Mouse IgG1-FITC (Cat. 555748), mouse IgG1-PE (Cat. 555749), mouse IgG2a-PE (Cat. 555574) and mouse IgG1-APC (Cat. 555751). Antibodies were titrated to find a concentration that gave optimum performance, and both antibodies and their isotype controls were used at the same concentration.

### 2.5. Cell Surface and Intracellular Staining

MNCs were first stained for surface antigen expression by incubation with the appropriate antibody or isotype control cocktails for 15 min at room temperature. After washing once with PBSAAE, stained cells were fixed and permeabilized (Fix and Perm kit, Life Technologies, Paisley, UK; GAS004) according to the manufacturer’s instructions. Anti-Bcl-2-FITC or hamster isotype-FITC was added to the permeabilized cells which were then left at room temperature in the dark for 45 min. Following one wash with PBSAAE, data were acquired on a Beckman Coulter Cytomics FC 500 MPL Flow Cytometry System and analyzed with the flow-cytometer integrated MXP software. Flow cytometry was also used for quantitative analysis of dendritic cell subsets. To this end, DC subset counts were performed in whole blood as described in [29] using a blood dendritic cell enumeration kit purchased from Miltenyi Biotec Ltd. (Bergisch Gladbach, Germany, Cat. 130-091-086), which included CD141/BDCA-3-APC (myeloid cDC) and CD303/BDCA-2-FITC for plasmacytoid DC (PDC). Red blood cells were lysed using a red cell lysing buffer (Sigma, Gillingham, UK; Cat. R7757) according to the manufacturer’s instructions. Just before data acquisition via flow cytometry (Beckman Coulter FC500 cytometer), 100 µL Flow-Count Fluorospheres (Beckman Coulter, High Wycombe, UK) were added to each tube to enable absolute counting.

### 2.6. Detection of sCD23 and Cytokine Levels

Frozen culture supernatants were thawed rapidly (at 37 °C) before analysis. sCD23 levels were measured using an enzyme-linked immunosorbent assay (Bender MedSystems, Vienna, Austria) and detected using a Biorad Benchmark Plus microplate spectrophotometer (Biorad, Watford, UK). Furthermore, cytokines were detected through multiplexed bead-array technology. A human Th1/Th2 10-Plex FlowCytomix™ kit was purchased from Bender MedSystems (Vienna, Austria, now eBioscience^®^) to detect IL-1β, IL-2, IL-4, IL-5, IL-6, IL-8, IL-10, IL-12p70, IFN-γ and TNF-α and used according to manufacturer’s instructions. Data were acquired on a Beckman Coulter FC500 flow cytometer and analyzed using MXP software (Beckman Coulter, High Wycombe, UK).

### 2.7. Statistics

A Linear ANOVA Model with three factors was first assayed. The factors were “type of allergy” with levels “none PN CME”, “Challenging agent” with levels “none PN CME” and “neutralizing antibody” with levels “anti-IL10, anti-IL4, anti-TGFb, anti-IL6 and none”. Prior to the analyses, the value of the measurements, V, was transformed into Ln(V + 1). Thereafter, the model was simplified as a two-way ANOVA model with two factors: “type of allergy” with levels “none PN CME” and “challenging agent” with levels “none PN CME”. The interaction between the effects of these two factors was not significant in any case. Least squares means were computed for each effect to estimate the significance of the differences between the levels of each factor. Canonical discriminant analyses were performed to find linear functions able to discriminate between non-allergic individuals and individuals allergic to CME or PN. Prior to analyses, each variable, V, was transformed as V′ = ln(V + 1). Cross-validation error did not increase. All statistical analyses were carried out using SAS 9.3 (SAS Institute Inc., Cary, NC, USA).

## 3. Results

### 3.1. Constitutive Plasma Cytokines in Food Allergy

In comparison with the plasma of non-allergic children, there were constitutively higher levels (*p* < 0.05) of IL-1β (Figure 1A), IL-2 (Figure 1B), IL-4 (Figure 1C), IL-5 (Figure 1D) and IL-10 (Figure 1F) in the plasma of CME and PN allergic children but lower levels of IL-12p70 (Figure 1G). There were no differences in plasma concentrations of IL-6 (Figure 1E), IFN-γ (Figure 1H) or TNF-α (Figure 1I) between allergic and non-allergic children. These data suggest either a predisposition towards Th2 responses in food allergy or on-going in vivo stimulation in affected subjects.

### 3.2. Spontaneous In Vitro sCD23 Release and Cytokine Secretion by MNC in Food Allergy

Consistent with the data above, MNCs from allergic children cultured for 6 days in the absence of added antigens spontaneously released significantly more (*p* < 0.05) sCD23 (Figure 2A), IL-1β (Figure 2B), IL-2 (Figure 2C), IL-4 (Figure 2D), IL-5 (Figure 2E), IL-10 (Figure 2F), IL-12p70 (Figure 2G), IFN-γ (Figure 2H) and TNF-α (Figure 2I), whereas MNC from non-allergic children secreted virtually none of the cytokines. 

### 3.3. Allergy-Associated Abnormalities in the Expression of Costimulatory Molecules

We examined the expression of CD80, CD83 and CD86 on in vitro unstimulated MNCs and those stimulated with PN or CME antigens. In 6-day unstimulated MNC cultures, there were significantly more (*p* < 0.05) CD80^+^ DR^−^ (Figure 3A), CD80^+^ CD83^+^ (Figure 3B), CD80^+^ CD86^−^ (Figure 3C) subsets in PN allergic than non-allergic MNCs and significantly less (*p* < 0.05) CD86^+^ CD80^−^ (Figure 3D) subsets in MNCs from both PN and CME allergic children than those from non-allergic children. There were no differences noted between PN and CME allergies. Antigenic stimulation did not alter the expression of costimulatory molecules. These data suggest dysregulated costimulatory molecule expression in the allergies, which could impact on allergenic responses in vivo.

### 3.4. Differences in the Numbers of Immune Cell Subsets

In order to assess if there was an imbalance of myeloid (mDC) and plasmacytoid dendritic cell (pDC) subsets as documented in atopic patients [30], we quantified absolute numbers of these subsets in whole blood of allergic and non-allergic children. There were significantly more (*p* < 0.05) mDC cells in the blood of CME allergic than either PN allergic or non-allergic children (Figure 4). Although the number of pDCs showed a similar trend, the differences did not reach statistical significance. These data highlight differences in DC subsets between PN and CME allergies that may have an impact on in vivo allergenic responses and resolution or lack of it.

We then checked whether the differences in mDC and pDC subsets were accompanied by changes in other cellular subsets. To this end, we quantified the percentages of B cells, T cells and APC following culture for 6 days in the absence or presence of PN or CME antigens. Irrespective of antigenic challenge, there were no differences in the percentages of B cells between the different study groups (Figure 5A). However, there were significantly more B cells in cultures from CME allergic children responding to PN antigens than CME antigens. T cell percentages were significantly higher (*p* < 0.05) in allergic than in non-allergic subjects (Figure 5B). This may be due to the spontaneous activation of T cells in food allergic subjects [8], resulting in their higher numbers. Antigenic challenge did not induce any changes. No differences were apparent in the percentages of APC between allergic and non-allergic children (Figure 5C).

### 3.5. Survival Advantage of Immune Cell Subtypes from Allergic Children

When the Bcl-2 content of cultured cells was examined, it seemed that the cells were not refractory to in vitro stimulation as previously thought. Taking this into consideration alongside the allergy-associated constitutive presence of cytokines in plasma and spontaneous release of soluble molecules during in vitro culture, we looked for the expression of CD69, a leukocyte activation molecule expressed in situations of chronic inflammation [31]. Furthermore, reduced levels of apoptosis were observed in a mouse model of food allergy. Importantly, cells escaping T cell-mediated apoptosis when transferred into a naïve animal triggered an antigen-specific IgE response suggesting an important role in the genesis of allergic reactions [24]. Thus, we also examined the expression of the pro-survival factor Bcl-2 to ascertain if MNCs from allergic children had a survival advantage. We report that the Bcl-2 content of CD69^+^ B cells (Figure 5D) and T cells (Figure 5E) was significantly higher (*p* < 0.05) than that of similar cells from non-allergic donors. The Bcl-2 content of CD69^+^ APC was comparable between the study subjects (Figure 5F). In all cases, challenge with CME or PN antigens did not alter the constitutive Bcl-2 expression (Figure 5E–G). These data suggest that activated T and B cells from allergic children have a survival advantage and that such longevity might influence the duration of a discrete allergic response or persistence of an individual’s allergic sensitivity over time.

The survivability of CD69^+^ T cells from food allergic children could be mediated through surface expression of CD152. It has been shown that signals induced by CD152 act directly on activated T lymphocytes to confer resistance to cell death mainly in Th2 cells [31,32,33]. While we found no differences in the expression of cell surface CD152 on CD4^+^ T cells from allergic and non-allergic children (Figure 6A), there was significantly higher expression (*p* < 0.05) on the surface of CD69^+^ CD4^+^ T cells from both CME and PN allergic compared with non-allergic donors (Figure 6B). These data suggest that in food allergy, CD152 may also confer longevity to activated CD4^+^ T cells.

### 3.6. IL-4-Mediated Regulation of CD152 Expression

As cell surface expression of CD152 can be upregulated by cytokines [34], we tested the ability of several neutralizing antibodies (anti-IL-4, -IL-6, -IL-10 and -TGF-β) to alter its expression. The only effect seen was on MNCs from PN allergic children challenged with their cognate allergen when neutralization of IL-4 brought cell surface CD152 expression down to the levels seen in non-allergic children (Figure 7). It has been reported that stimulation in the presence of IL-4 induces CD152 expression in B cells [35], and this may also be the case with T cells.

### 3.7. Canonical Discriminant Analysis

Canonical discriminant analyses were performed to find linear functions able to differentiate between non allergic individuals and individuals allergic to CME or PN. Only subjects with complete data sets have been included in these analyses. Linear discriminant functions based on plasma measurements of in vivo (Table 1) and in vitro (Table 2) cytokine secretion were efficient in discriminating between non-allergic and allergic individuals but not between individuals allergic to PN or CME (Figure 8A,B). Nevertheless, there was some separation between CME and PN allergic children as well as overlap between the two allergies reflecting their similarities, differences and variations in the severity of their particular allergies.

## 4. Discussion

This study suggests that children with CME and PN allergy diverged in their mechanisms of immune regulation/dysregulation, and most importantly, both populations displayed the survival advantage of key cell types. We show the constitutive presence of Th2, pro-inflammatory and regulatory cytokines in the plasma of allergic children. With regard to regulatory cytokines, IL-10 and TGF-ß are found mainly in the allergen-specific Treg cell population [36], and there is strong evidence that peripheral T-cell regulation has a crucial role in the control of allergies. Our data are in agreement with a previous observation by [37] with regard to IL-4, but the allergic children in our study were older and had virtually no IL-12p70 with higher levels of IL-10, indicative of in vivo Th2 bias. IL-4 has been shown to enhance IL-10 gene expression in Th2 cells [38]. This could represent negative feedback regulation, except that plasma IL-10 levels are comparable between PN tolerant and PN allergic individuals [39]. Furthermore, severe acute anaphylaxis episodes can occur in allergic patients with high IL-10 levels [40]. Indeed, several studies have shown that IL-10 may promote the development of Th2 cells [41,42,43] and facilitate IgE production [44,45] to contribute to allergy-related disease through collaboration with IL-1β [46], which was also higher in our allergic subjects. There is a requirement for IL-1 in Th2 cell activation [47], and IL-1β can induce the production of IL-2 [48], which was also higher in the plasma of our allergic children. In turn, IL-2 may be responsible for the increased plasma IL-5 in our allergic subjects [49], and successful oral immunotherapy for peanut allergy is associated with a decrease in both IL-5 and IL-2 [50]. IL-5 induces terminal maturation of eosinophils, prolongs eosinophil survival by delaying apoptotic death, possesses eosinophil chemotactic activity, increases eosinophil adhesion to endothelial cells and enhances eosinophil effector functions [51]. So, its presence in plasma suggests an enhanced ability to mount Type I hypersensitivity responses. This poses an important question as to why these cytokines are present in the circulation of allergic children who have avoided eating the offending foods. The answer may lie, at least partly, in molecular cross-reactivity of food allergens with other plant food sources or with aeroallergens in the environment [52,53,54] because most food allergic children have other chronic atopic conditions such as asthma, eczema or allergic rhinitis.

The constitutive presence of Th2 cytokines in the plasma of allergic children was accompanied by the spontaneous release of these and other cytokines in MNC cultures. A previous study [8] showed that in the absence of allergen exposure, Th2 cytokine secretion by T cells from food allergic donors is DC dependent. In agreement with our study, they found that no further increase in these cytokines occurred after stimulation with allergens and concluded that T cells from these subjects were already activated in vivo rendering them unresponsive to additional allergenic stimulation in vitro. They found abnormalities in costimulatory molecule expression in eosinophilic esophagitis but not in food allergy. In our study, however, using a different detection panel, we did find differences in costimulatory molecule expression between non-allergic versus allergic and PN versus CME allergies. The differences from non-allergic children involved CD80^+^ subsets with a lower expression of other costimulatory molecules as well. Due to the complexities of costimulatory interactions, it is impossible to speculate on the impact of these changes on immune responsiveness. However, it is recognized that aside from overlapping functions, CD80 and CD86 have distinct functional identities. CD80 is associated with diminution of immune responses through interaction with its functional ligand CD152, to exert tolerogenic effects when expressed by immature dendritic cells in the absence of CD86 upregulation [55]. Certainly, there were lower percentages of CD80 and CD86 double positive cells following in vitro challenge with either CME or PN antigens. These unusual subsets in our allergic children may impair the normal maturation of dendritic cells that has been described in allergy [56]. It is also noteworthy that IL-10 was constitutively high in the plasma of our allergic children, and its presence during DC maturation impairs the upregulation of costimulatory molecules [32]. On a speculative note, in the presence of such allergic immune dysregulation weakly immunogenic food allergens may pose a problem. It is equally possible that the failure to upregulate CD86 may be due to in vivo activation rendering the cells refractory to further stimulation. One could argue that in vivo activation should yield a higher proportion of CD86^+^ cells, but this may not be so as CD86 has faster association and dissociation rates than CD80 [55] making stable complex formation unlikely. CD152-CD80 regulatory interactions are likely to proceed in the absence of CD86. Indeed, CD69^+^ cells that were present in our allergic participants had more CD152 expressed.

Signals generated by CD152 on activated CD69^+^ T cells induce resistance to cell death [23]. This notion has a direct bearing on our observation that in allergic children, levels of the anti-apoptotic Bcl-2 within CD69^+^ CD152^+^ T cells were significantly higher. Dysregulation of apoptosis, a cell death pathway key for the development and the maintenance of homeostasis, is at the core of a variety of diseases [57]. Most importantly, we have previously observed that in a mouse model of food allergy, a lack of apoptosis occurred during antigen presentation [24] and that the adoptive transfer of apoptosis-resistant cells triggered an antigen-specific response in naïve mice even in the absence of antigen administration [58]. Further, the added presence of IL-4, also a viability factor, would serve to sustain high Bcl-2 levels within both B and T cells [22,59].

It is important to highlight that to date the presence of cells displaying a clear survival advantage in food allergy has not been reported in humans. The longevity of allergic effector T cells could influence both the magnitude and duration of responses amplified by a higher frequency of allergen-specific T cells as seen in IBD [60]. We hypothesize that expanded populations of allergen-specific T cells are maintained by on-going stimulation in vivo through cross reaction with other food proteins and that these cells contribute to exaggerated Type I hypersensitivity responses through resistance to the intrinsic cell death pathway.

Despite the Th2 bias of both CME and PN allergies, IL-4-mediated regulation of CD152 on the surface of CD4^+^ T cells after challenge with PN was seen in PN and NA donors but not in CME donors. Other differences included sCD23 and IL-1β release, costimulatory molecule expression, DC1 subsets, and a greater survival advantage of CD69^+^ B cells from PN allergic donors. In view of the plasticity of DCs, the impact of higher numbers in CME allergy is not known. Canonical discriminant analysis confirmed the distinction between PN and CME allergies. We suggest that aside from the greater allergenicity of seed storage proteins found in peanuts, there are inherent differences in the regulation of PN versus other food allergies in affected individuals.

## 5. Conclusions

In summary, this study suggests the existence of differences in various immunological parameters between different allergies and non-allergic children, the most noticeable being the survival advantage acquired in allergic individuals, through higher Bcl-2 expression in T and B cells. To date, the latter was observed in a mouse model of food allergy [24] but was not described in humans. We are aware of the small number of patients utilized in this study. Thus, the results enable us to formulate the hypothesis that the lack of apoptosis is pivotal to the development of food allergy, which will need to be confirmed by a study involving a larger cohort of subjects. The latter will also enable the investigation on the mechanism underlying the allergy-related survival advantage of certain cell types key to the regulation of immune response.

## Figures and Tables

**Figure 1 cells-12-02736-f001:**
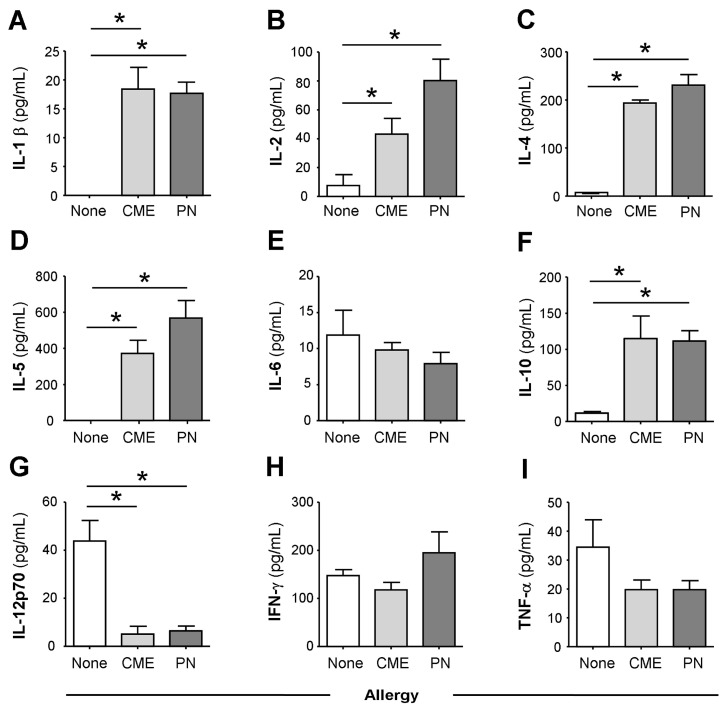
Constitutive plasma cytokines. (**A**) IL-1β, (**B**) IL-2, (**C**) IL-4, (**D**) IL-5, (**E**) IL-6, (**F**) IL-10, (**G**) IL-12p70, (**H**) IFN-γ and (**I**) TNF-α were evaluated using multiplex bead technology in the plasma of non-allergic children and those with CME or PN allergies. Data were acquired via flow cytometry and are shown as means ± s.e.m; * *p* < 0.05, analyzed using ANOVA with adjustment for multiple comparisons using Tukey.

**Figure 2 cells-12-02736-f002:**
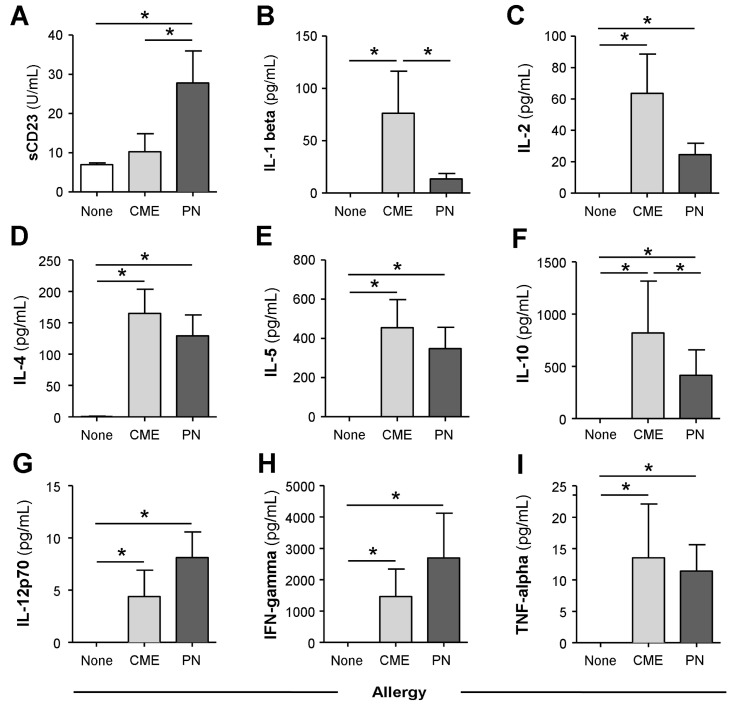
Spontaneous in vitro sCD23 and cytokine secretion. MNCs from non-allergic children or those with CME or PN allergies were cultured for six days without allergenic stimulation. Culture supernatants were collected and their levels of (**A**) sCD23 were detected using ELISA, while cytokines (**B**) IL-1β, (**C**) IL-2, (**D**) IL-4, (**E**) IL-5, (**F**) IL-10, (**G**) IL-12p70, (**H**) IFN-γ and (**I**) TNF-α were measured using multiplex bead technology and flow cytometry. Data are shown as means ± s.e.m; * *p* < 0.05, analyzed using ANOVA with adjustment for multiple comparisons using Tukey.

**Figure 3 cells-12-02736-f003:**
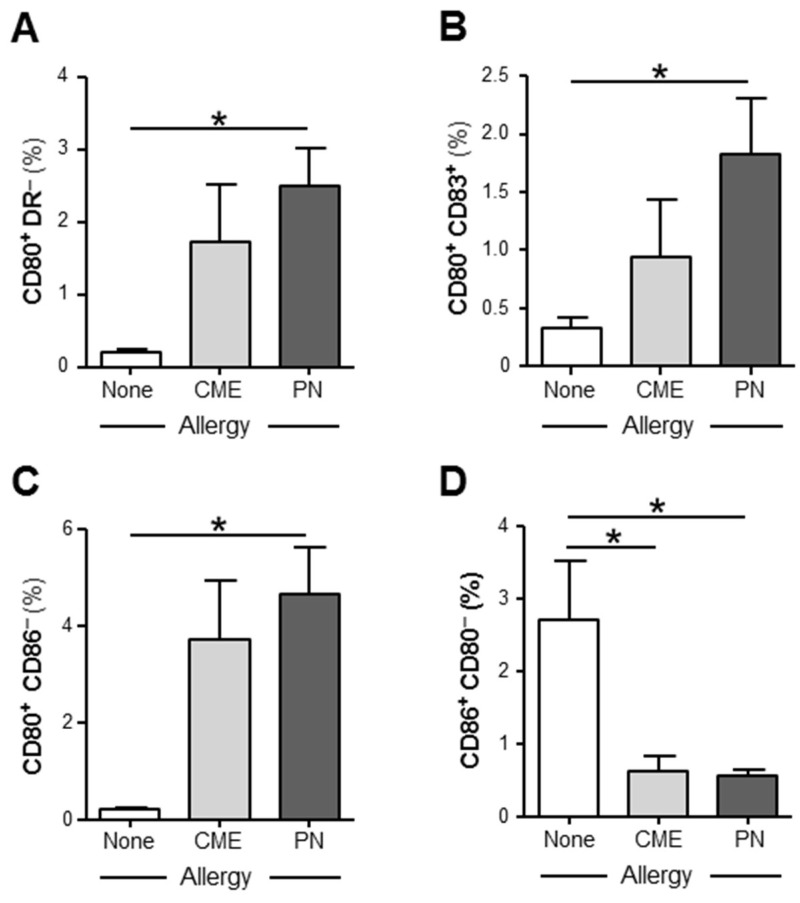
Expression of costimulatory molecules. MNCs from non-allergic children and those with CME or PN allergies were cultured without allergenic stimulation. On day 6, cells were harvested, washed and stained with CD80-FITC, CD83-PE, CD86-APC and HLA-DR-ECD. Data for the percentages of (**A**) CD80^+^HLA DR^−^, (**B**) CD80^+^ CD83^+^, (**C**) CD80^+^ CD86^−^ and (**D**) CD86^+^ CD80^−^ were acquired via flow cytometry and are shown as means ± s.e.m; * *p* < 0.05, analyzed using ANOVA with adjustment for multiple comparisons using Tukey.

**Figure 4 cells-12-02736-f004:**
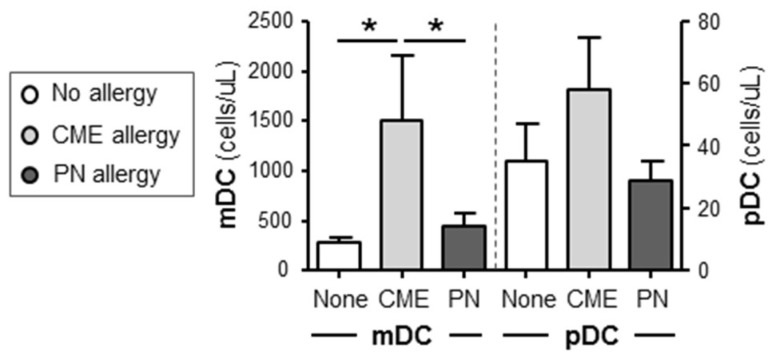
Dendritic cell subset enumeration. Dendritic cells were quantified in heparinized whole blood from non-allergic children and those with CME or PN allergies. Data were acquired via flow cytometry and are shown as means ± s.e.m; * *p* < 0.05, analyzed using ANOVA with adjustment for multiple comparisons using Tukey.

**Figure 5 cells-12-02736-f005:**
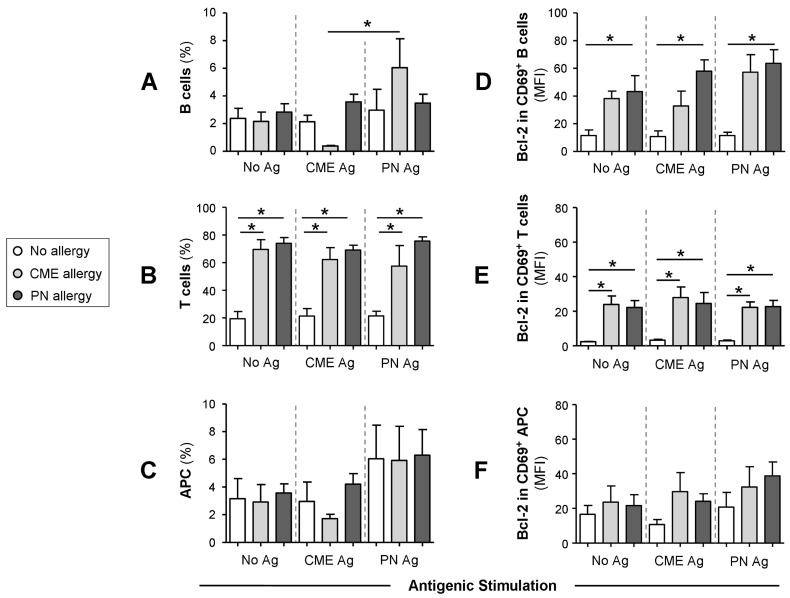
Percentages of B cells, T cells, APCs and their Bcl-2 content. MNCs from non-allergic children and those with CME or PN allergies were first stained with each of two panels for cell surface antigens, both containing CD69-APC, CD80-FITC, CD83-FITC and CD86-FITC except that Panel 1 had CD3-ECD and Panel 2 had CD19-ECD as lineage marker. This was followed by intracellular staining with Bcl-2-PE. Percentages of (**A**) B cells, (**B**) T cells and (**C**) APC are shown alongside MFI values for Bcl-2 content within CD69^+^ (**D**) B cells, (**E**) T cells and (**F**) APC. All data were acquired via flow cytometry and are seen as means ± s.e.m; * *p* < 0.05, analyzed using ANOVA with adjustment for multiple comparisons using Tukey.

**Figure 6 cells-12-02736-f006:**
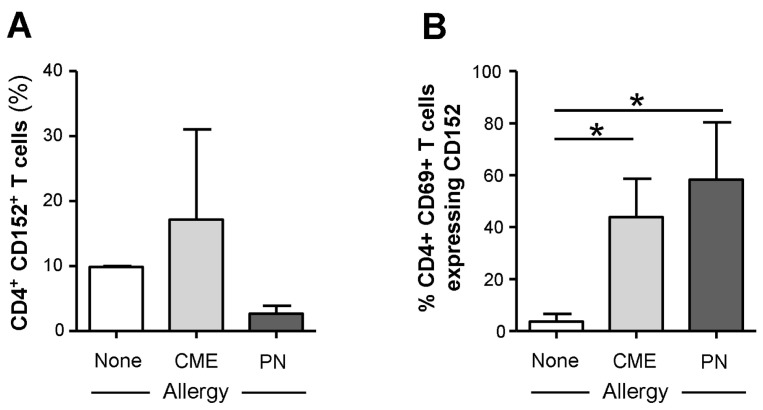
Cell surface expression of CD152 within CD69^+^ CD4^+^ T cells. MNCs from non-allergic children and those with CME or PN allergies were stained with CD3-FITC, CD152-PE, CD4-ECD and CD69-APC. Cells were gated using scatter profiles and CD3 expression to identify T cells. Data for the percentages of CD152^+^ cells within (**A**) CD4^+^ or (**B**) CD4^+^ CD69^+^ were acquired via flow cytometry and are shown as means ± s.e.m; * *p* < 0.05, analyzed using ANOVA with adjustment for multiple comparisons using Tukey.

**Figure 7 cells-12-02736-f007:**
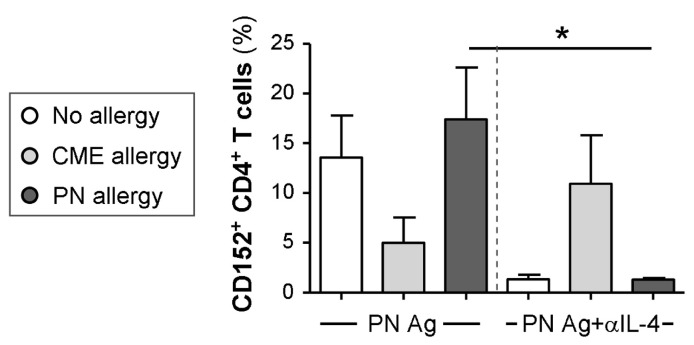
Cytokine-mediated regulation of CD152 expression. MNCs from non-allergic children and those with CME or PN allergies were cultured in the presence or absence of PN antigens (Ag) and with or without an IL-4 neutralizing antibody. On day 6, cells were harvested, washed and stained with CD3-FITC, CD152-PE and CD4-ECD. Data for the percentages of positive cells were acquired via flow cytometry and are shown as means ± s.e.m; * *p* < 0.05, analyzed using ANOVA with adjustment for multiple comparisons using Tukey.

**Figure 8 cells-12-02736-f008:**
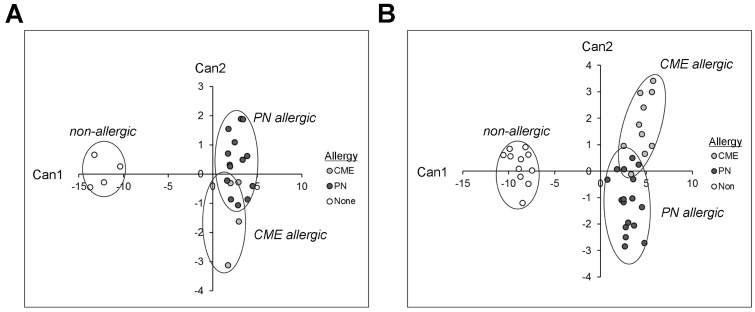
Canonical coefficients of non-allergic individuals and individuals allergic to CME or PN. Linear discriminant functions were based on cytokines secreted (**A**) in vivo and therefore constitutively present in plasma or (**B**) released following 6-day MNC culture in vitro.

**Table 1 cells-12-02736-t001:** Linear discriminant functions based on in vivo plasma cytokines.

Variable	No Allergy	CME Allergy	PN Allergy
Constant	−67.44	−198.85	−218.99
*Ln IL-2	9.66	14.39	16.71
Ln IL-4	39.86	66.13	69.06
Ln IL-5	−9.99	8.74	9.78
Ln IL-1β	−19.71	−10.74	−13.80
Ln IL-10	4.98	−4.57	−5.22
Ln IL-12p70	5.93	−2.47	−1.79

*Ln = natural logarithm.

**Table 2 cells-12-02736-t002:** Linear discriminant functions based on in vitro cytokine release.

Variable	No Allergy	CME Allergy	PN Allergy
Constant	−0.35	−19.39	−16.52
*Ln IL-2	−0.01	1.33	1.29
Ln IL-4	0.33	1.41	1.57
Ln IL-5	0.05	3.99	3.92
Ln IFN-γ	0.26	−0.65	−0.63
Ln IL-10	−0.18	2.19	1.46
Ln IL-12p70	−0.16	−3.22	−2.76

*Ln = natural logarithm.

## Data Availability

Data are available upon request.

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
