# Peer review of "Alteration of Immunoregulatory Patterns and Survival Advantage of Key Cell Types in Food Allergic Children"

_cells, 2023, doi:10.3390/cells12232736_

Round 1
Reviewer 1 Report
Comments and Suggestions for Authors
General comment: The authors studied the immunoregulatory patterns and survival of different immune cells in children allergic to peanut and milk. The experimental design, analytical techniques, and data analysis were appropriate. The authors did a thorough investigation. The manuscript in general is well written and thoroughly discussed. I have only a few minor comments and suggestions in my specific comments.
Specific comments:
Line 121: Gating strategies should be provided.
Line 150: Add commas between different types of allergy and challenging agents.
Lines 154-155: Same as above.
Line 161: Which post-hoc test was used?
Line 175: “ANOVA”.
Line 309: “importantly”?
Lines 314-315: Awkward sentence.
Line 370: “importantly”?
Line 371: Awkward sentence.
Line 379: What is IB?
Line 392: The authors should not expand the conclusion to other food allergies.
Comments on the Quality of English LanguageThe manuscript in general is well written. There are a few grammatical and syntax errors.
Reviewer 2 Report
Comments and Suggestions for Authors
This is an investigation into cytokines and immune cells in children with food allergy. Children with peanut and cow’s milk allergy are observed to have unique cytokine signatures and cell activation markers as compared to healthy controls.
The findings are interesting but I have some comments and concerns.
1) There is inadequate information about the clinical aspects. Given the striking findings we need to know more about the allergic and control children. Onset of allergy, current diet, other atopic conditions, etc. IgE and total IgE levels would be helpful.
2) The findings in Fig 1 are striking but also quite surprising given that these are constitutive levels and do not involve stimulation. I don’t see an exact parallel of this in the literature, but I don’t think other studies show this striking difference in an antigen non-specific manner
3) Has viability been assessed at day 6? The absolute lack of any signal in the control children begs the question of whether the cells are alive. Were there any non-antigen positive controls (eg, LPS or PHS, or IL-2/28, etc).
4) Showing flow cytometry gates in Fig 3 AND Fig 5 would help with concerns about the cells. Relatedly, what are all the cells in the controls in fig 5 if they are not T or B cells?
5) Fig 2, 5, 6, 7 all build off each other, but again, a simple explanation is that cell viability is poor in unstimulated controls. Simple proliferation studies would likely have been as helpful as the data I am looking at.
6) Line 47 and 51: Are CD28/152-CD80/86 actually crucial to allergic reactions or just to sensitization?
Comments on the Quality of English Languagesome comments were provided above
Reviewer 3 Report
Comments and Suggestions for Authors
This is a clinical study investigating the role of immune alteration on the prognosis of food allergy, milk, egg, and peanut allergy, in the children with atopic dermatitis. This study provides an important evidence suggesting that Bcl-2 content in B cells is higher in the children with peanut allergy than that in the children with milk/egg allergy, of which hypothesis arose from the investigation using mouse model. Although the result was obtained with relatively small number of children, this result is worth being confirmed and extended further by following investigation.
